# MRI of the Entire Spinal Cord—Worth the While or Waste of Time? A Retrospective Study of 74 Patients with Multiple Sclerosis

**DOI:** 10.3390/diagnostics11081424

**Published:** 2021-08-06

**Authors:** Esben Nyborg Poulsen, Anna Olsson, Stefan Gustavsen, Annika Reynberg Langkilde, Annette Bang Oturai, Jonathan Frederik Carlsen

**Affiliations:** 1Department of Radiology, Rigshospitalet, Copenhagen University Hospital, 2100 Copenhagen, Denmark; esbennyborg@hotmail.com (E.N.P.); Annika.Langkilde@regionh.dk (A.R.L.); 2Danish Multiple Sclerosis Center, Department of Neurology, Copenhagen University Hospital, Rigshospitalet, 2600 Glostrup, Denmark; anna.gabriella.olsson@regionh.dk (A.O.); stefan.gustavsen@regionh.dk (S.G.); annette.oturai@regionh.dk (A.B.O.); 3Department of Clinical Medicine, University of Copenhagen, 2200 Copenhagen, Denmark

**Keywords:** MRI, multiple sclerosis, spinal cord, dissemination in time, dissemination in space

## Abstract

Spinal cord lesions are included in the diagnosis of multiple sclerosis (MS), yet spinal cord MRI is not mandatory for diagnosis according to the latest revisions of the McDonald Criteria. We investigated the distribution of spinal cord lesions in MS patients and examined how it influences the fulfillment of the 2017 McDonald Criteria. Seventy-four patients with relapsing-remitting MS were examined with brain and entire spinal cord MRI. Sixty-five patients received contrast. The number and anatomical location of MS lesions were assessed along with the Expanded Disability Status Scale (EDSS). A Chi-square test, Fischer’s exact test, and one-sided McNemar’s test were used to test distributions. MS lesions were distributed throughout the spinal cord. Diagnosis of dissemination in space (DIS) was increased from 58/74 (78.4%) to 67/74 (90.5%) when adding cervical spinal cord MRI to brain MRI alone (*p* = 0.004). Diagnosis of dissemination in time (DIT) was not significantly increased when adding entire spinal cord MRI to brain MRI alone (*p* = 0.04). There was no association between the number of spinal cord lesions and the EDSS score (*p* = 0.71). MS lesions are present throughout the spinal cord, and spinal cord MRI may play an important role in the diagnosis and follow-up of MS patients.

## 1. Introduction

Multiple sclerosis (MS) is an inflammatory demyelinating disease affecting both the brain and the spinal cord [1,2]. MS is the most frequent cause of disability in the young and middle-aged population in developed countries [3]. Upliftingly, growing evidence suggests that early diagnosis and treatment may delay or even prevent disability in MS patients [4]. With the promise of effective disease control, fiscal societal costs for treating MS have increased steadily since the 1990s [5,6]. The costs for treating patients with severe disabilities from MS especially are mounting. Though generally well-tolerated, new disease-modifying therapies do carry risks of complications and warrant repeated monitoring of patients [7,8]. Accordingly, clinicians face the task of balancing the efficacy of treatment against safety and costs. To that end, reliable diagnostic tools are essential in procuring prompt and correct diagnosis and to secure optimal treatment evaluation.

Magnetic resonance imaging (MRI) is a reliable diagnostic tool and readily identifies MS involvement of both the brain and the spinal cord [9]. The diagnosis of MS is achieved with a combination of clinical work-up, spinal fluid analysis, and MRI as described in the McDonald Criteria first published in 2001 and revised several times since, most recently in 2017 [10,11]. MRI is used both to establish whether the diagnostic criteria of dissemination in space (DIS) and time (DIT) are fulfilled and to monitor patients during treatment. Hailing from times when MRI scanners were scarce and scan time was very limited, MS scan protocols are often reduced to a brain scan with a single field-of-view (FOV) of the spinal cord, roughly covering the upper two thirds of the latter. In line with this, international guidelines consider brain and cervical spine MRI sufficient for most MS patients, leaving thoracic and lumbar spinal cord MRI for patients with spinal cord symptoms and for patients that do not fulfill DIS criteria otherwise [12]. Adding a second FOV of the lower medulla would depending on the sequences used, prolong the entire scan time by as much as 15–20 min. This prolonged acquisition time may lead to discomfort and claustrophobia to the patients resulting in low diagnostic quality or premature termination of the MRI scans [13]. On the other hand, scanning the entire spinal cord may yield a more certain diagnosis and provide a better point of reference for later treatment evaluation.

The primary aim of this study was to describe the spatial distribution of lesions in the entire spinal cord in MS patients. Secondary aims were to examine how entire spinal cord MRI influences the fulfillment of the 2017 McDonald Criteria for DIS and DIT and to investigate whether the number of spinal cord lesions relates to disability.

## 2. Materials and Methods

### 2.1. Patients

This study was approved by the regional ethics committee and the Danish Data Protection Agency (protocol no.:H-16047666). Written informed consent forms were obtained from all participants. The study was performed as an observational study with consecutive, prospective inclusion of patients and retrospective evaluation of data. From 2017 to 2019 79 patients were prospectively and consecutively included at the Danish Multiple Sclerosis Center, Copenhagen University Hospital, Denmark (Figure 1). Inclusion criteria were newly diagnosed patients (<1 year) with treatment naïve relapsing-remitting MS (RRMS) or clinically isolated syndrome (CIS), age between 18 and 55, and fulfillment of the 2017 McDonald criteria [11]. Exclusion criteria were other autoimmune disorders, celiac disease, lactose intolerance, other severe diseases (e.g., cancer, vasculitis, HIV, hepatitis, inflammatory bowel disease), prior treatment with chemotherapy, pregnancy and breastfeeding, and treatment with corticosteroids or antibiotics during the last 4 weeks. The cohort has previously been included and described in other studies [14,15].

No later than two months from inclusion patients had MRI of both the brain and the entire spinal cord (C1-L2) performed before and after gadolinium (Gd) contrast administration. At the same time point, a neurologic exam was performed by one of two medical doctors, and disability was evaluated using the Expanded Disability Status Scale (EDSS) [16]. Data from all patients including age, sex, and MS subtype were obtained at the time of MRI and EDSS scoring.

### 2.2. MRI Protocol

Brain and spinal cord MRI was performed on a Siemens 3T Verio scanner (Siemens, Erlangen, Germany). All brain scans were performed with either a 32- or 64-channel head/neck matrix). The MRI protocol of the brain included a sagittal/3D T1 Magnetization Prepared—Rapid Gradient Echo (MPRAGE) (repetition time (TR): 1900 ms, echo time (TE): 2.26 ms, inversion time (TI): 900 ms, slice gap: 0.5 mm, matrix: 256 × 256, FOV: 256 mm, voxel dimensions: 1.0 × 1.0 × 1.0 mm, K-Space filling: linear), a sagittal/3D T2 Fluid-attenuated inversion recovery (FLAIR) (TR: 5000 ms, TE: 395 ms, TI: 1800 ms, slice gap: 0 mm, matrix 256 × 256, FOV: 250 mm, voxel dimensions: 1.0 × 1.0 × 1.0 mm), a transversal T2 TSE (TR: 6000 ms, TE: 87 ms, slice gap: 1 mm, slice thickness: 3 mm, FOV: 220 mm, voxel dimensions: 0.7 × 0.7 × 3 mm) and sagittal/3D T1 MPRAGE after intravenous Gd contrast injection with the same scan parameters as before Gd.

The MRI of the spinal cord was made using a spine coil and scanned with two FOVs. The protocol for the cervicothoracic spinal cord included a sagittal T1 Turbo Spin Echo (TSE) after Gd contrast injection (TR: 400 ms, TE: 11 ms, pixel dimensions: 0.9 × 0.9 mm, slice thickness: 2.5 mm, slice gap: 0.25 mm, matrix: 288 × 320, FOV: 275 mm) a sagittal T2 (TR: 3000 ms, TE: 103 mm, pixel dimensions: 0.7 × 0.7 mm, slice thickness: 2.5 mm, slice gap: 0.25 mm, matrix: 288 × 384, FOV: 275 mm), a sagittal T2 Turbo inversion recovery magnitude (TIRM) (TR: 3000 ms, TE: 57 ms, TI: 200 ms, pixel dimensions: 0.7 × 0.7 mm, slice thickness: 2.5 mm, slice gap: 0.25 mm, matrix: 192 × 384; FOV: 275 mm) and transversal T2 of lesions (TR: 3500 ms, TE: 107 ms, voxel dimensions: 0.6 × 0.6 mm, slice thickness: 5 mm, slice gap: 0.5 mm, FOV: 200 mm).

The protocol for the thoracolumbar spine included a sagittal T1 Turbo Spin Echo (TSE) after Gd contrast injection (TR: 400 ms, TE: 11 ms, pixel dimensions: 0.9 × 0.9 mm, slice thickness: 2.5 mm, slice gap: 0.25 mm, matrix: 307 × 384, FOV: 350 mm) a sagittal T2 (TR: 3000 ms, TE: 103 mm, pixel dimensions: 0.9 × 0.9 mm, slice thickness: 2.5 mm, slice gap: 0.25 mm, matrix: 288 × 384, FOV: 275 mm), a sagittal T2 TIRM (TR: 3000 ms, TE: 57 ms, TI: 200 ms, pixel dimensions: 0.9 × 0.9 mm, slice thickness: 2.5 mm, slice gap: 0.25 mm, matrix: 192 × 384; FOV: 275 mm) and transversal T2 of lesions with the same parameters as for the cervicothoracic spinal cord.

The intravenous Gd-infusion used was 0.1 mmol/kg Multihance^®^ [gadobenate dimeglumine] (Bracco International B.V., Amsterdam, The Netherlands).

### 2.3. MRI Data Analysis

All MR images were retrospectively analyzed by one neuroradiologist with 15 years of experience in MRI (ARL). The reader was blinded to clinical data. All scans were presented to the reader in the local PACS system (Impax 6.7, Agfa, Mortsel, Belgium). Cerebral lesions were identified on T2 and TIRM images and their location within the brain was determined (periventricular, infratentorial, and juxtacortical/cortical). Contrast-enhancing lesions were also identified and quantified. Spinal cord involvement was identified on sagittal T2 and TIRM images and the level of segment involvement was recorded along with an assessment of contrast-enhancement. Multiple lesions at one spinal cord level were not registered separately but instead registered as one level spinal cord involvement. If lesions spanned through multiple segments spinal cord involvement was recorded separately for all segments.

Following the 2017 McDonald criteria, DIS was established if there were ≥1 T2-hyperintense lesion in two of four of the following locations: Periventricular, infratentorial, juxtacortical/cortical, and spinal cord. The number DIT was established if there was simultaneous presence of enhancing and non-enhancing lesions [5].

### 2.4. Statistical Analysis

SPSS 25.0 (IBM, Armonk, NY, USA) was used for statistical analyses. Chi-square tests were used to analyze the distribution between categorical scores. Fisher’s exact test was used if the expected number in any group was <5. The changes in the number of patients diagnosed with DIS when scanning the brain alone, the brain and the cervical spinal cord, the spinal cord contained within one FOV (C1-T8), and the entire spinal cord were examined using a one-tailed McNemar’s test. The change in numbers of patients diagnosed with DIT was also examined using a one-tailed McNemar’s test. A *p*-value < 0.05 was considered statically significant except for the one-sided McNemar’s test, where a *p*-value of <0.025 was considered significant. Tables and figures were created with Microsoft Office Excel 2011.

## 3. Results

### 3.1. Demographics and Clinical Findings

Overall, the cohort included 79 patients diagnosed with RRMS or CIS. Five patients did not have complete spinal cord MRI performed and were consequently excluded from analyses. Two of the spinal cord MRI scans were performed six months later than brain MRI but showed no change from previous (pre-study) spinal cord MRI and were included. Nine patients did not receive contrast for either brain or spinal cord evaluation and were excluded from DIT analyses (Figure 1).

In total, 74 patients were included for data analysis with an average age of 35 years (range: 21–53, SD = 8.12) and a female to male ratio of 3.11. Seventy-two patients had RRMS and two had CIS. The median EDSS score was 2 (range: 0–4).

### 3.2. Brain MRI

T2 brain lesions were present in all patients and median EDSS scores according to the number of lesions and lesion locations are presented in Table 1. Of the 65 patients who received contrast injection as part of their brain MRI, 26.2% (17/65) had contrast-enhancing lesions. A total of 60 contrast-enhancing lesions were found in all the included patients.

### 3.3. The Spinal Cord

Spinal cord lesions were present in 87.8% (65/74) of patients (Figure 2 and Figure 3). In all, 257 spinal cord lesions were recorded. The distribution of spine lesions is shown in Figure 3A. The average fraction of segments involved pr. patient in the cervical and thoracic part was 0.24 and 0.15 respectively. Figure 3B shows the distribution of patients according to spinal cord segment involvement. Both the distribution of spinal cord lesions and the number of patients with lesions at each level showed a bimodal distribution with peaks at the lower cervical and thoracolumbar regions. Of the 65 patients who had MRI after contrast, 16.9% (11/65) had contrast-enhancing lesions. Seventeen contrast-enhancing lesions were found in the spinal cord, 47.0% (8/17) were found in the cervical part, and the remaining 53.0% (9/17) were found in the thoracic spinal cord. No contrast-enhancing lesions were seen in the lumbar spinal cord. Of the 11 patients with contrast-enhancing lesions, 54.5% (6/11) had lesions in the cervical spine and 72.7% (8/11) had lesions in the thoracic spinal cord.

### 3.4. Dissemination in Space and Time

Based on the evaluation of brain MRI scans alone 58 of 74 patients fulfilled the DIS criteria (Figure 4A). When evaluating the cervical spinal cord in addition to brain MRI, 67 patients fulfilled the DIS criteria, which was significantly more than with brain MRI alone (*p* = 0.004). When evaluating the entire spinal cord DIS was fulfilled in 71 patients, which was not significantly more than when scanning the brain and cervical spine (*p* = 0.067). There was no significant difference between scanning C1-T8 (correlating with one FOV), which yielded DIS in 70 patients and scanning the entire spinal cord (*p* = 0.5).

Based on cerebral MRI alone 17 of 65 patients fulfilled the DIT criteria as compared to 19 of 65 when evaluating the cervical spinal cord in addition to brain MRI alone (*p* = 0.48) (Figure 4B). When evaluating the entire spinal cord DIT was fulfilled in 23 patients, which was not significantly more than when scanning the brain and cervical spine (*p* = 0.13), or when scanning the brain alone (*p* = 0.04). There was no statistically significant difference between scanning C1-T8, which yielded DIT in 22 patients and scanning the entire spinal cord (*p* = 1.0).

### 3.5. EDSS

Stratifications of the number of spinal cord lesions and EDSS scores are shown in Figure 5. The median EDSS scores of the patients within each interval of lesions were: 1.5 (0–2 lesions), 2 (3–5 lesions), 2 (6–8 lesions), 2.75 (9–11 lesions) and 2.5 (12–15 lesions). There was no statistically significant difference between the EDSS score for these groups (*p* = 0.710).

## 4. Discussion

This is the first paper to examine the impact of performing MRI of the entire spinal cord in MS diagnosis, following the latest revision of the McDonald criteria, i.e., the 2017 McDonald criteria [11]. We demonstrated that demyelinating lesions were present throughout the spinal cord with a bimodal distribution comprising a cervical and a thoracolumbar peak. Cervical spinal cord MRI had an impact on DIS evaluation while scanning the entire spinal cord did not increase the number of patients that fulfilled the DIS criteria. For fulfillment of DIT criteria, there was no significant change when scanning the spinal cord, as compared to scanning the brain only.

Our findings of a bimodal distribution of spinal cord lesions are in agreement with previous studies [17,18,19,20,21,22]. However, many of these studies use scan protocols that do not consequently include the entire medulla or do not describe the FOV, anatomical range, or MRI scanners used [18,19,20,21,22]. It has been speculated that the bimodal distribution of lesions is due to the varying diameter of the spinal cord, and consequently the amount of myelin, in both the cervical and thoracolumbar region [22,23].

Two previous studies have examined the impact of adding spinal cord MRI in addition to brain MRI in the diagnosis of MS following the earlier versions of the McDonald criteria [20,24]. Though both studies find some added effect on the fulfillment of the DIS criteria, neither of the studies performed an entire spinal cord MRI. Jacobi et al. found no added effect of spinal cord MRI on the fulfillment of DIT [24]. The present study is the first study on entire spinal cord MRI and the first to examine the fulfillment of DIS and DIT criteria after the newest McDonald criteria. In accordance with these earlier studies, we found an increased number of patients fulfilling the DIS criteria when adding cervical spinal cord MRI to brain MRI, as suggested by current guidelines. We only saw a trend towards an increased number of patients with DIS when scanning the entire spinal cord, as compared to scanning the brain and cervical spinal cord only. Likewise, we only found a trend towards increased fulfillment of DIT when scanning the entire spinal cord as compared with brain MRI alone. An encouragement for performing spinal cord MRI was seen in a recent study showing that spinal cord MRI is often more accurately assessed than brain MRI in MS diagnosis and often contributes more to the diagnosis than do brain lesions [25].

Once the diagnosis of MS is established, MRI remains the most important paraclinical tool for evaluating disease progression and treatment response [12]. Studies have shown that due to clinically silent lesions, the number of lesions on MRI is four to twelve times larger than the number of clinical relapses during the same period [19]. This is, in combination with the high number of spinal cord lesions in the lower medulla, a valid argument for performing an entire spinal cord MRI to have as elaborate a baseline scan as possible. With the advent of new disease-modifying therapies promising to further postpone or even prevent clinical relapse in MS patients, even small changes in the para-clinical workup may become crucial to secure optimal treatment [4]. To that end, a thorough MRI examination including whole spinal cord imaging may become key to monitor disease progression as it could warrant a change in therapy before the advent of clinical symptoms.

There was no association between the number of spinal cord lesions and the EDSS score in the present study. Other studies have shown a positive association between EDSS score and the number of spinal cord lesions in MS patients [26,27]. However, the first of the above-mentioned studies only investigated the cervical region, and the second study only evaluated diffuse abnormalities in the spinal cord. In general, it is believed that spinal cord lesions are more likely to be disabling to patients than brain lesions, and our findings may be explained by a small sample size, newly diagnosed patients with low EDDS scores, and primarily RRMS phenotype. Further, a recent study has shown that the total volume of spinal cord lesions rather than the number of lesions is correlated with the EDSS score [28].

This study had both strengths and limitations. Among its strengths is the uniform MRI methodology used covering the entire spinal cord and the assessments of MRI scans performed by the same experienced neuroradiologist. Further, patient inclusion followed strict criteria, and MRI and clinical scorings were performed at predefined time points. A limitation of the study is the relatively small number of patients included, and that the patients participating were mainly diagnosed with RRMS, potentially excluding the extrapolation of results to other MS phenotypes. Axial spinal cord images were acquired post-hoc on lesions detected on the sagittal images, and not on the entire spinal cord, which is a limitation of the study’s sensitivity. Additionally, counting and including every spinal cord segment in lesions spanning through multiple segments and denoting multiple lesions on a segment as one, might have affected the results. The EDSS scores in our patient cohort were generally low with a small range, as patients were newly diagnosed. The correlation between spinal cord involvement and EDSS score would possibly be better evaluated on a group of patients with more varied EDSS scores.

## 5. Conclusions

In conclusion, this paper displays the broad spatial distribution of spinal cord lesions in newly diagnosed MS patients. We have confirmed that scanning the spinal cord may play a role in fulfilling DIS criteria for some patients with the newest revisions of the McDonald criteria. The impact of entire spinal cord MRI in the diagnosis and follow-up of MS patients should be further examined in larger studies.

## Figures and Tables

**Figure 1 diagnostics-11-01424-f001:**
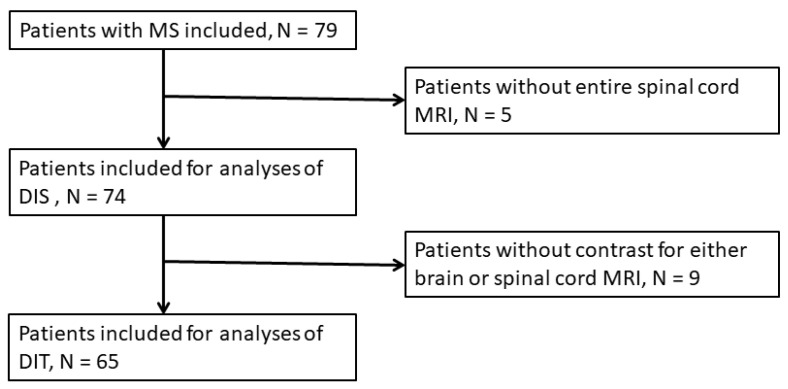
Study flowchart.

**Figure 2 diagnostics-11-01424-f002:**
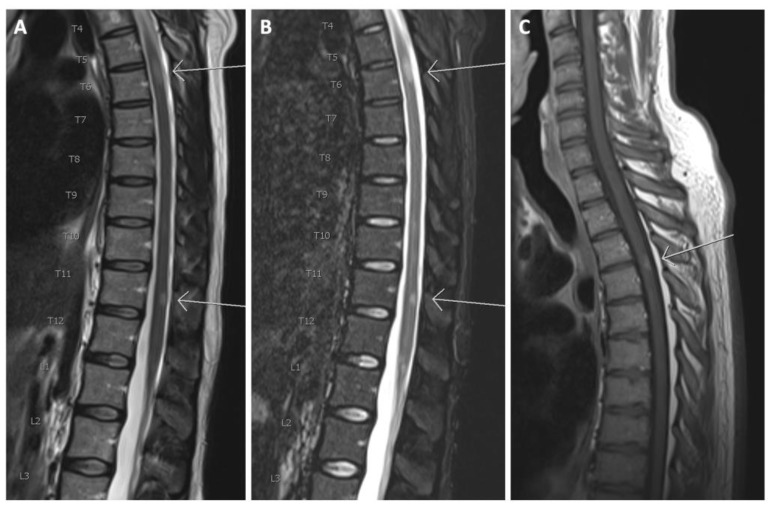
(**A)** Sagittal T2 and (**B**) sagittal TIRM of the thoracolumbar spinal cord. Arrows show T2 hyperintense lesions at levels T6 and T11; (**C**) T1 after gadolinium in another patient. The arrow shows a contrast-enhancing lesion at level T3/T4.

**Figure 3 diagnostics-11-01424-f003:**
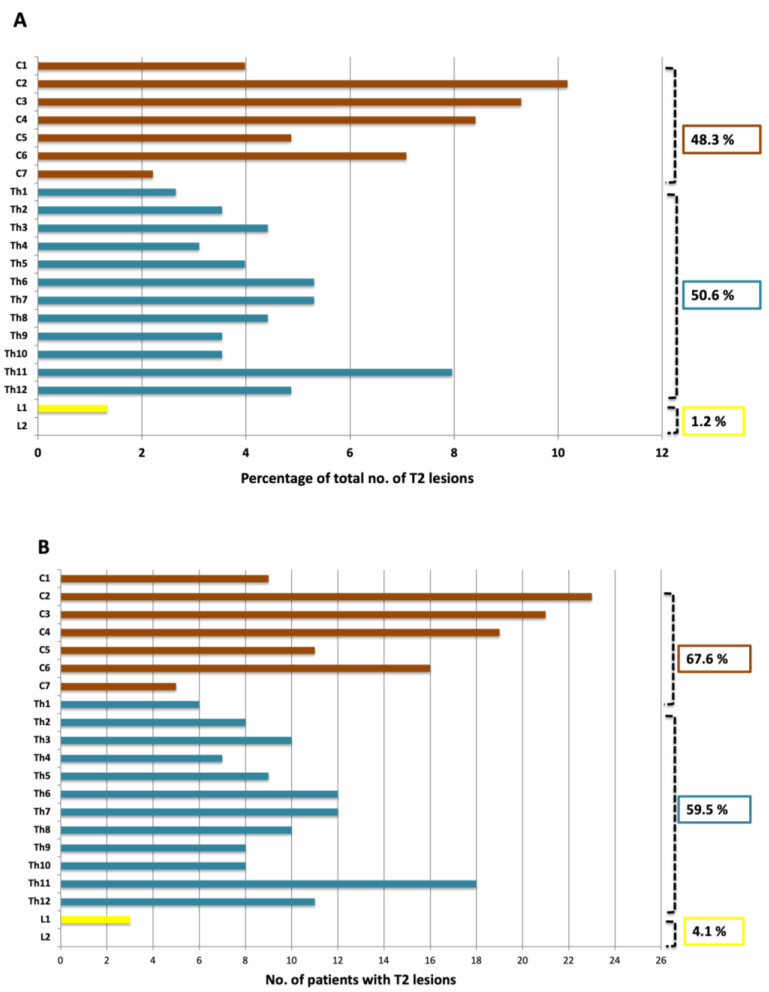
(**A**) Segmental distribution of T2 lesions in the cervical, thoracic and lumbar spine; (**B**) Distribution of patients with lesions in the cervical, thoracic and lumbar spine.

**Figure 4 diagnostics-11-01424-f004:**
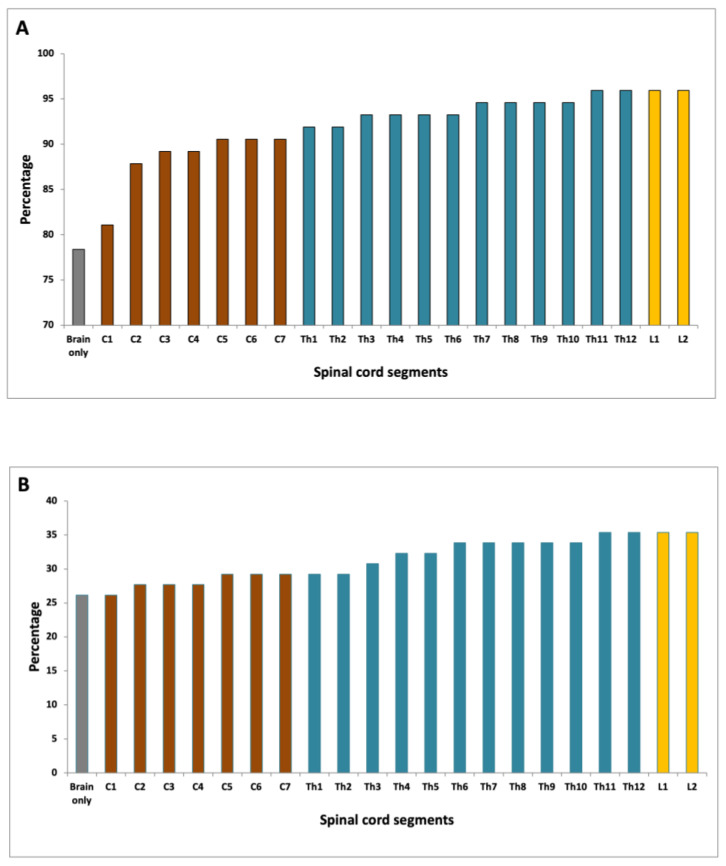
(**A**) Fulfillment of DIS according to the spinal cord level evaluated; (**B**) Fulfilment of DIT according to the spinal cord level evaluated.

**Figure 5 diagnostics-11-01424-f005:**
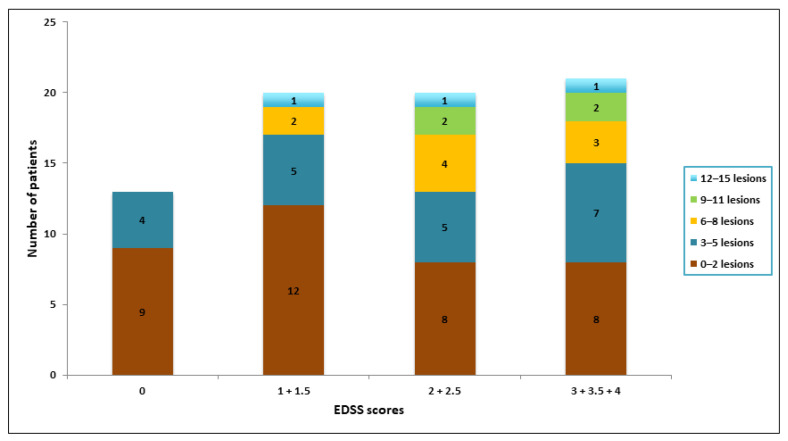
Bar diagram showing categorizations of lesion number pr. patient and grouped EDDS scores.

**Table 1 diagnostics-11-01424-t001:** Results from brain MRI and corresponding EDSS scores.

T2 Lesions			
	Patients, *n*	Pct. of Total	Median EDSS Score
**Total Number of Patients:**	74	100	2
**No. of lesions:**			
<10	23	31.0	1.5
10–20	21	28.4	2
>20	30	40.5	2.5
**Location:**			
Periventricular	72	97.3	2
Infratentorial	51	68.0	2
Juxtacortical/cortical	44	59.5	2
Gadolinium-enhancing lesions			
	**Patients, *n***	**Pct. of total**	
**Patients receiving contrast (brain + spinal cord):**	65	-	
**Patients with enhancing lesions:**	17	26.2	
**Number of lesions**			
**Gadolinium-enhancing lesions:**	60	-	

## Data Availability

The data presented in this study are available on request from the corresponding author. The data are not publicly available due to local legislation.

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
