# Peer review of "MRI of the Entire Spinal Cord—Worth the While or Waste of Time? A Retrospective Study of 74 Patients with Multiple Sclerosis"

_diagnostics, 2021, doi:10.3390/diagnostics11081424_

Round 1

Reviewer 1 Report

the authors analyzed the impact of entire spine MRI on fulfilling 2017 McDonalds criteria.

The study is interesting and adds substantial new information in a field rich of interest.

I have some minor concerns:

1) page 2 line 66: study design should be better described, patient were prospectively included in the MS center, but the study design appears to be an observational retrospective study on consecutive MS patients

2) page 3 line 121: a single tailed McNemar's test appears appropriate because a whole spinal cord MRI can only show more lesions with respect to brain or brain + cervical MRI, but with a single tailed test the appropriate level of significance should be 0.025

3) In the discussion session it should be interesting to highlight the importance of a whole spinal cord MRI as a baseline reference in order to monitor disease evolution and treatment response

Author Response

We would like to thank the reviewer for a thorough revision and constructive comments. We have revised our manuscript after reading the reviewer’s comments and find that the manuscript has improved. Below is our point by point reply to the reviewer’s comments.

Reviewer 1

The authors analyzed the impact of entire spine MRI on fulfilling 2017 McDonalds criteria.

The study is interesting and adds substantial new information in a field rich of interest.

I have some minor concerns:

1) page 2 line 66: study design should be better described, patient were prospectively included in the MS center, but the study design appears to be an observational retrospective study on consecutive MS patients

We agree with the reviewer. An addendum has been made to the manuscript. (Page 2, M&M, section 2.1, first paragraph).

2) page 3 line 121: a single tailed McNemar's test appears appropriate because a whole spinal cord MRI can only show more lesions with respect to brain or brain + cervical MRI, but with a single tailed test the appropriate level of significance should be 0.025

We thank the reviewer for pointing this out and have made corrections in the manuscript. (Page 3, M&M, section 2.4, first paragraph and page 7, Results, section 3.4, second paragraph ). Further, as this change has an impact on the interpretation of the results, the Abstract, Discussion (page 9, third paragraph) and Conclusions have been altered accordingly.

3) In the discussion session it should be interesting to highlight the importance of a whole spinal cord MRI as a baseline reference in order to monitor disease evolution and treatment response.

We agree with the reviewer. We have added a discussion of the possibly gains of doing whole spinal cord MRI as baseline reference for treatment response evaluation (page 10, Discussion, fourth paragraph).

Reviewer 2 Report

In this paper, the Authors described the prevalence of demyelinating lesions in the cervical, cervical+thoracic (Th1-Th8) and whole spinal cord, and tested their value for DIS and DIT determination. Patients’ disability assessed by using EDSS was also compared between category groups stratified by lesion number.

I have several comments and recommendations the Authors may consider:

Introduction and rationale:

- In my opinion, the statement “On the other hand, scanning the entire spinal cord will allow clinicians to make the best treatment decisions[…]” should not be included in the introduction because it anticipates conclusion that should be eventually drawn based on the study results. Furthermore, I’m confused because it appears from the Authors results thereafter, that the contribution from the thoracic spinal cord rings a non-significant contribution to the DIS and DTI criteria fulfillment. Please clarify and rephrase.

-The “extended” FOV covering the cervical + Th1-Th8 is somewhat arbitrary because it represents a local center choice.

Methods

-The MRI protocol details are insufficient. Please provide information about matrix and/or voxel size, gap. The “sagittal T2 TIRM Turbo Inversion recovery SPACE” is an unusual technique for brain MS imaging: is this a FLAIR (Dark Fluid) sequence? Is this a short-tau sequence? Please spell out acronyms, specify the 3D k-space filling, and provide the TI value. What is “MPR”? MPRAGE? Other?

-Why were different Gd contrast agent doses employed (0.1 to 0.2 mmol/kg)? How were they determined?

-Was the reader blinded to clinical data? How were images presented? PACS? Which software?

-“Contrast-enhancing lesions were also identified and quantified”: were quantitative analyses (e.g. volume, CNR) performed? Please specify “quantified” or rephrase.

-“Multiple lesions of one spinal cord level, was not registered separately” Please clarify/rephrase.

-As for statistics, in my opinion, there more informative methods which might be employed to test the impact of the spinal cord coverage protocol choice on patients’ DIS/DIT and MS diagnosis probability, such as likelihood (OR) and/or binomial regression analyses, that the Authors might consider. Correlation analyses might be also performed for EDSS.

Discussion

-It seems that the main novelty item of the present work would be related to the assessment of the entire spinal cord compared to the cervical spinal cord only. However, only a trend was found for the contribution to the DIS criteria fulfillment. There was no difference for DIT. These are basically negative findings, even though the Authors encourage the use of the entire spinal cord imaging in the end. This is not clear to me. I suggest the Authors to provide some additional analyses to support their results, and to discuss more clearly if and why they recommend, or not, to dedicate additional scan time to the dorsal cord.

-Even though most previous studies failed to demonstrate a relationship between EDSS and spinal cord lesions, it is worth discussing previous findings from [Pravatà et al, MSJ 2019], showing that a relationship is present if the lesion volume rather than number is estimated.

-Limitations: Axial images were acquired post-hoc on lesions detected on the sagittal images, and not on the entire spinal cord. This limits the study sensitivity, and should be stated.

There are some English language syntax errors.

Author Response

We would like to thank the reviewer for a thorough revision and constructive comments. We have revised our manuscript after reading the reviewer’s comments and find that the manuscript has improved. Below is our point by point reply to the reviewer’s comments.

Reviewer 2

In this paper, the Authors described the prevalence of demyelinating lesions in the cervical, cervical+thoracic (Th1-Th8) and whole spinal cord, and tested their value for DIS and DIT determination. Patients’ disability assessed by using EDSS was also compared between category groups stratified by lesion number.

I have several comments and recommendations the Authors may consider:

Introduction and rationale:

- In my opinion, the statement “On the other hand, scanning the entire spinal cord will allow clinicians to make the best treatment decisions[…]” should not be included in the introduction because it anticipates conclusion that should be eventually drawn based on the study results. Furthermore, I’m confused because it appears from the Authors results thereafter, that the contribution from the thoracic spinal cord rings a non-significant contribution to the DIS and DTI criteria fulfillment. Please clarify and rephrase.

We agree with the reviewer, that the phrasing anticipates the results of the study. We have rephrased to emphasize the uncertainty of the statement. Further, we have down toned the recommendation of whole spinal cord MRI in the abstract, discussion and conclusion to a more moderate statement.

-The “extended” FOV covering the cervical + Th1-Th8 is somewhat arbitrary because it represents a local center choice.

We agree with the reviewer that the definition of the extended FOV is arbitrary. In our search of literature on the topic, the extend of the FOVs used are very poorly described, and most studies published do not even mention, if one or two FOVs are used, if the use of one or two FOVS are used consistently etc, which level of the spinal cord is covered as a minimum etc. Accordingly, we find, that the approach we have used is at least well described, explainable and reproducible. 

Methods

-The MRI protocol details are insufficient. Please provide information about matrix and/or voxel size, gap. The “sagittal T2 TIRM Turbo Inversion recovery SPACE” is an unusual technique for brain MS imaging: is this a FLAIR (Dark Fluid) sequence? Is this a short-tau sequence? Please spell out acronyms, specify the 3D k-space filling, and provide the TI value. What is “MPR”? MPRAGE? Other?

We thank the reviewer for the comment. The additional information has been added to the manuscript (Page 2, M&M, section 2.2, first paragraph).

-Why were different Gd contrast agent doses employed (0.1 to 0.2 mmol/kg)? How were they determined?

We thank the reviewer for pointing this out. All scans were performed using a 0.1 mmol/kg dose. This has been corrected in the manuscript (Page 3, M&M, section 2.2., second paragraph)

-Was the reader blinded to clinical data? How were images presented? PACS? Which software?

We thank the reviewer for the comment. The reader was blinded to clinical data. MRI scans were presented in the local PACS software, IMPAX 6.7 (Agfa, Mortsel, Belgium). This information has been added to the manuscript  (Page 3, M&M, section 2.3., first paragraph).

-“Contrast-enhancing lesions were also identified and quantified”: were quantitative analyses (e.g. volume, CNR) performed? Please specify “quantified” or rephrase.

We agree with the reviewer, that this description is inadequate. No volumetric or other quantitative analyses of contrast-enhancing lesions were performed. We have rephrased the paragraph concerning contrast-enhancement (Page 3, M&M, section 2.3., first paragraph).

-“Multiple lesions of one spinal cord level, was not registered separately” Please clarify/rephrase.

We agree with the reviewer, that this description is unclear. Multiple lesions at one spinal cord level were not registered separately but instead registered as one level of involvement. This has been clarified in the manuscript (Page 3, M&M, section 2.3., first paragraph).

-As for statistics, in my opinion, there more informative methods which might be employed to test the impact of the spinal cord coverage protocol choice on patients’ DIS/DIT and MS diagnosis probability, such as likelihood (OR) and/or binomial regression analyses, that the Authors might consider. Correlation analyses might be also performed for EDSS.

We agree with the reviewer, that more informative statistical methods would be desirable. However, we are not sure how to perform a binomial regression analyses, as the dependent variable in such an analysis would have to be the DIS/DIS status of patients with brain and entire spinal cord MRI. The status of both brain MRI and different FOVs of the spinal cord would without doubt be associated with the DIS/DIT status and there would be a significant interaction between the all the independent variables. Concerning EDDS score and correlation with the number of spinal cord lesions, we find that this could be interesting, but at the same time realize that our data concerning only newly-diagnosed MS patients with generally low EDSS scores with a small range would probably not fit such an analysis. We have added a paragraph to our limitations concerning the limited range of the EDDS in our patient cohort (page 10, discussion, fifth and sixth paragraph).    

Discussion

-It seems that the main novelty item of the present work would be related to the assessment of the entire spinal cord compared to the cervical spinal cord only. However, only a trend was found for the contribution to the DIS criteria fulfillment. There was no difference for DIT. These are basically negative findings, even though the Authors encourage the use of the entire spinal cord imaging in the end. This is not clear to me. I suggest the Authors to provide some additional analyses to support their results, and to discuss more clearly if and why they recommend, or not, to dedicate additional scan time to the dorsal cord.

We agree with the reviewer, that the recommendation in the manuscript extended beyond what was warranted by the results. As mentioned above, we have down toned the recommendation of whole spinal cord MRI in the abstract, discussion and conclusion to a more moderate statement.

-Even though most previous studies failed to demonstrate a relationship between EDSS and spinal cord lesions, it is worth discussing previous findings from [Pravatà et al, MSJ 2019], showing that a relationship is present if the lesion volume rather than number is estimated.

We thank the reviewer for drawing our attention to this study. We have added a line in the discussion concerning this interesting finding and have cited the above-mentioned article.

-Limitations: Axial images were acquired post-hoc on lesions detected on the sagittal images, and not on the entire spinal cord. This limits the study sensitivity, and should be stated.

We agree with the reviewer. The manuscript has been altered accordingly (page 10, discussion, last paragraph).

There are some English language syntax errors.

We thank the reviewer for the comment. The manuscript has been proofread after revision.

Round 2

Reviewer 2 Report

-